# Inositol-Exchange Activity in Human Primordial Placenta

**DOI:** 10.3390/ijms25063436

**Published:** 2024-03-19

**Authors:** Bence Géza Kovács, Gergely Asbóth, Dorina Supák, Balázs Mészáros, Tamás Marton, Nándor Ács, Sándor Valent, Zoltán Kukor

**Affiliations:** 1Department of Obstetrics and Gynecology, Semmelweis University, 1082 Budapest, Hungary; kovacs.bence@semmelweis.hu (B.G.K.); meszaros.balazs@semmelweis.hu (B.M.); marton.tamas@semmelweis.hu (T.M.); acs.nandor@semmelweis.hu (N.Á.); 2Institute of Biochemistry and Molecular Biology, Semmelweis University, 1082 Budapest, Hungary; asboth.gergely@med.semmelweis-univ.hu (G.A.); kukor.zoltan@semmelweis.hu (Z.K.); 3Department of Pathology, Forensic and Insurance Medicine, Semmelweis University, 1082 Budapest, Hungary

**Keywords:** human placenta, placentation, trophoblast, inositol, phosphatidylinositol, phosphatidylinositol synthase, phosphatidylinositol exchange enzyme

## Abstract

Human placenta is an intensively growing tissue. Phosphatidylinositol (PI) and its derivatives are part of the signaling pathway in the regulation of trophoblast cell differentiation. There are two different enzymes that take part in the direct PI synthesis: phosphatidylinositol synthase (PIS) and inositol exchange enzyme (IE). The presence of PIS is known in the human placenta, but IE activity has not been documented before. In our study, we describe the physiological properties of the two enzymes in vitro. PIS and IE were studied in different Mn^2+^ and Mg^2+^ concentrations that enabled us to separate the individual enzyme activities. Enzyme activity was measured by incorporation of 3[H]inositol in human primordial placenta tissue or microsomes. Optimal PIS activity was achieved between 0.5 and 2.0 mM Mn^2+^ concentration, but higher concentrations inhibit enzyme activity. In the presence of Mg^2+^, the enzyme activity increases continuously up to a concentration of 100 mM. PIS was inhibited by nucleoside di- and tri-phosphates. PI production increases between 0.1 and 10 mM Mn^2+^ concentration. The incorporation of [3H]inositol into PI increased by 57% when adding stabile GTP analog. The described novel pathway of inositol synthesis may provide an additional therapeutic approach of inositol supplementation before and during pregnancy.

## 1. Introduction

The placenta is a highly invasive structure derived from the conceptus shortly after implantation, having two main components, trophoblasts and extraembryonic mesoderm. Extravillous trophoblast cells have high migratory and invasive properties [1,2,3,4,5]. The mass of the placenta grows to 500 g throughout pregnancy. The growth and the migration are facilitated and governed by growth factors. A number of growth factors are produced by trophoblasts, including EGFs, CSF1, TGF-α, TGF-ß, IGF, PlGF, VEGF, erythropoietin and PDGF-like protein [6]. The robust proliferative capacity of trophoblasts is facilitated by growth factor receptors that have been identified in the trophoblast: such as insulin receptor, insulin-like growth factor receptor (IGF)-IR, EGF receptor, corticotrophin-releasing factor receptor, hepatocyte growth factor receptor, erythropoietin receptor, VEGF receptors, granulocyte–macrophage colony-stimulating factor receptor, CFS receptor, PDGF receptor and ERBB2. The signal transduction network of these receptors acts via phosphatidylinositol (PI) and its polyphosphate derivatives (PIP_x_). It is not surprising that the phospholipids of the human placenta have considerable PI content. In the early placenta, 6.9% of phospholipids are PI, compared to 5.5% in term placentae [7]. 

A number of different phosphatidylinositol derivates are involved in signal transduction. Two of the most important are phosphatidylinositol(4,5)-diphosphate and phosphatidylinositol(3,4,5)-triphosphate (PIP3). PIP3 is a membrane lipid that acts as a docking site for proteins possessing the pleckstrin homology domain. Such proteins are important regulators of cell survival, proliferation and differentiation. RAC-alpha serine/threonine protein kinase is an important pleckstrin homology (PH) domain protein, and its activity is required for normal preimplantation embryo development and survival [8]. VEGF promotes angiogenesis, and its effect is mediated by phosphoinositide-specific phospholipase C (PLC) activation, releasing inositol 1,4,5-triphosphate and elevating Ca^2+^ level. The other product, diacylglycerol, activates PKC, and it has an important role in the growth signal transduction [9]. Phosphatidylinositols are usually esterified with arachidonic acid in the *sn*-2 position. About 40% of the long chain polyunsaturated fatty acids are bound in PI [10].

PIP_x_s are versatile, having a role in the regulation of cell growing, proliferation, differentiation and apoptosis. PI and PIP_x_ may have a role in the ATP-induced trophoblast hormone secretion and active transport of nutrients too [11].

Direct PI synthesis is possible in two distinct ways: through phosphatidylinositol synthase (PIS) and through inositol exchange enzyme (IE) [12,13]. Pathways of phosphatidylinositol synthesis are shown in Figure 1. PIS is extensively studied in the context of phosphatidyl inositol metabolism; however, IE is usually overlooked [14]. PIS catalyzes the CDP-DAG + inositol → PI + CMP reaction. CDP-DAG is synthetized by the CDP-diacylglycerol synthetase enzyme from phosphatidic acid and CTP. Human placental PIS was first isolated and characterized by Antonsson in 1994. The placental human PIS is typically a microsome-bonded enzyme. It has an optimum pH of 9.0 and requires Mn^2+^ or Mg^2+^ to function, but it is inhibited by Ca^2+^ or Zn^2+^ ions. Optimal activity is obtained between 0.5 and 2.0 mM Mn^2+^ concentration, but higher concentrations inhibit the enzyme activity. In the presence of Mg^2+^, the enzyme activity increases gradually with the increasing Mg^2+^ ion concentration, peaking at 100 mM Mg^2+^. The maximal activity in the presence of Mg^2+^ is approximately four times higher than the maximal activity that is obtained with Mn^2+^ alone. It is of note that adding 2 mM Mn^2+^ does not affect the Mg^2+^ activation [13]. PIS is inhibited by nucleoside di- and tri-phosphates [15].

In contrast to PIS, cytosol-based IE catalyzes the *phosphatidyl-X + inositol → PI + X* reaction. In this process, “X” can represent different substrates such as choline or ethanolamine. The IE enzyme’s main function is remodeling of existing phospholipids. In a different experiment, activity of IE, extracted from rat liver, similarly to PIS, is inhibited by 1 mM Ca^2+^ but is only slightly stimulated in 10 mM Mg^2+^. Within a 2 mM Mn^2+^ solution, however, there was a marked stimulatory effect. IE enzyme activity is augmented by guanosine 5‘-3-O-(thio)triphosphate (GTP-γ-S) and slowly hydrolysable guanine analogues of GTP with Mn^2+^. However, a combination of GTP-γ-S and Mg^2+^ decreases the enzyme activity, whilst GTP-γ-S with Ca^2+^ has no effect [16]. Tóth and Hertelendy described IE activity in rat uterus. They demonstrated that incorporation of inositol into PI is activated markedly and dose dependently with Mn^2+^ (0.1–10 mM) but on the contrary is inhibited by Ca^2+^ (1–10 mM) [12].

The third enzyme playing a role in PI synthesis is the phosphatidylinositol-inositol exchange enzyme (PI-IE). This represents an alternative pathway for inositol incorporation, but this enzyme does not participate in the net synthesis of PI (PI + inositol → inositol + PI). The enzyme was first characterized by Bleasdale and Wallis, who isolated it from rabbit lung [17]. 

There are similarities in how PIS and PI-IE are regulated. Both enzyme activities require Mn^2+^ or Mg^2+^. In both, optimal enzyme activity is achieved at concentrations of Mn^2+^ not greater than 1 mM, and when Mg^2+^ is added, concentrations greater than 10 mM are required for optimal enzyme activity. Similarly to PIS and IE, PI-IE activity is also inhibited by Ca^2+^, but in a less sensitive way, decreasing only by 12% in 1 mM Ca^2+^ (PIS is more sensitive to Ca^2+^ inhibition, with 63% decrease in activity in similar Ca^2+^ concentration). PI-IE activity requires presence of cytidine nucleotides, such as CMP, dCMP, CDP or CTP. In the absence of added nucleotides, there is no enzyme activity [18]. 

Antonsson isolated PIS from the human placenta and examined the activity of the purified enzyme [14] but they did not measure IE activity. Based on the work of Tóth and Hertelendy, it is likely that the human trophoblast can also produce IE [13]. We managed to isolate IE in the crude membrane fraction from the trophoblasts of the human placenta and characterized its biochemical properties. 

## 2. Results

It is known that the human placenta has very intensive PI synthesis. We aimed to isolate and measure IE activity in human primordial placenta. The experiments were conducted in the absence of CTP and CDP-DAG, ensuring that PIS and PI-IE activity were minimal.

### 2.1. Intracellular Localization of IE Activity

Microsomes, crude membrane (CM) and cytosol fraction were incubated in Ca^2+^- and PO_4_^3−^-free Hank solution for 60 min with 10 mM Mn^2+^ (IE activity) and 2.5 µCi [^3^H]inositol at 37 °C. We measured the radioactivity of PI. The microsome specific activity was 6.5-fold higher than CM activity (*p* < 0.01). In contrast, the cytosol fraction did not have significant radioactivity. The total labeling PI distribution was 45% in microsomes and 55% in CM (Figure 2). 

### 2.2. Effect of Ca^2+^ in PI Synthesis

Removal of endogenous Ca^2+^ from the tissue by ionophore A23187 and 5 mM EGTA did not result in significant change in the incorporation of myo-[3H]inositol into phosphatidylinositol in 10 mM Mn^2+^. This finding indicates that, irrespective of the IE origin, phosphatidylinositol synthesis was not inhibited by endogenous Ca^2+^ concentrations.

### 2.3. Characterization of the Role of IE Activity in the PI Production of Trophoblast Tissue

The minced human trophoblast PI production is Mn^2+^- and Mg^2+^-dependent. To measure the enzyme activity in relation to Mn^2+^ concentration, the trophoblast tissue was incubated with [3H]inositol in Ca^2+^- and PO_4_^3−^-free Hank solution at 37 °C for 60 min, and the radioactivity of PI was measured. At lower concentrations of Mn^2+^, until 1 mM, there was a sharp increase in the incorporation of radiolabeled inositol with further gradual rise until 10 mM of Mn^2+^ concentration (Figure 3). The [3H]inositol incorporation in PI is a time-dependent reaction between 1 mM and 10 mM Mn^2+^ at 37 °C up to 180 min. Despite the fact that the activity of PIS decreases above 2 mM Mn^2+^ [13], in our experiment, the radioactivity of PI increased up to 5 mM Mn^2+^, indicating that the increasing PI level has to be a result of something other than PIS activity.

### 2.4. Effect of GTP/GIDP in PI Synthesis 

PIS activity is inhibited by nucleoside di-,triphosphates. We examined the effect of GTP, GIDP (guanylyl-imido-diphosphate; stabile GTP analogue) and GDP on [3H]inositol incorporation in PI. Nucleotides express their activity only in the presence of Mg^2+^, so we used Mg^2+^ solution in our experiment (Figure 4). All nucleotides enhanced [3H]inositol incorporation, but the increase was only significant in GIDP (GIDP 57% vs. GTP 30% and GDP 11%).

### 2.5. Effect of Mg^2+^ and Mn^2+^ on Microsomal PI Synthesis

The radioactivity of [^3^H] PI in MS was increased in 10 mM Mg^2+^ or 10 mM Mn^2+^ during the 60 min incubation time, but the radioactivity of [^3^H] PI was 4 times higher in 10 mM Mn^2+^ than in 10 mM Mg^2+^ (Figure 5A,B). When we combined 10 mM Mn^2+^ with Mg^2+^ in the solution, in the presence of 1 mM Mg^2+^, the radioactivity of [^3^H] PI was only slightly elevated, but with 10 mM Mg^2+^, the radioactivity of [^3^H] PI decreased (Figure 5C). A concentration of 1 mM Ca^2+^ decreased the radioactivity of microsomal PI. *n* = 6 ± SD.

## 3. Discussion

A significant part of the human primordial placenta cell membrane is phosphatidylinositol derivates. They play a role in the growth of cells, proliferation, differentiation and apoptosis as well.

Placental PIS was identified in 1994, and now we identified IE activity in primordial human placenta. Previously, it was documented that purified PIS enzyme activity is decreased above 2 mM Mn^2+^ concentration [13]. In our experiment, we detected an increased radioactivity of [^3^H] PI in minced human trophoblast products and in MS. We demonstrated that, in contrast to the characteristics of PIS, there was a dose-dependent activity with Mn^2+^ concentrations between 0.1 and 10 mM. Our result suggests that it is IE, and not PIS, that is responsible for the increased activity of [^3^H] PI in the primordial placenta. Nucleoside-di- and trisphosphate decrease the activity of PIS but increase the activity of IE [15]. Supporting our hypothesis that we observed true IE activity, in our experiment, we measured higher [^3^H] PI radioactivity in the presence of GTP and GIDP (stabile derivate) than in the control group. We postulate that IE is responsible for this increase. Removal of endogenous Ca^2+^ from the tissue increased the incorporation of [3H]inositol into phosphatidylinositol by the CDP-diglyceride:inositol transferase enzyme in aorta slices [19]. In contrast, in our experiment, the endogenous Ca^2+^ did not have the same effect on the activity of IE. 

Our observations suggest that IE might have a much greater importance in PI synthesis in physiological conditions than previously thought. We measured the PI production of MS in the presence of Mg^2^. The PI synthesis was elevated by increasing Mg^2+^ concentrations [0–10 mM]. In the absence of CTP and CDP-DAG, the PIS activity was minimal in our experiments, but we render that the PIS activity is primary in the primordial placental phosphatidylinositol synthesis. It was documented previously that the maximal activity of purified PIS was four times higher in the presence of Mg^2+^ in contrast to Mn^2+^. This was mirrored in our experiment: the radioactivity of [^3^H] PI was four times higher in 10 mM Mn^2+^ than in 10 mM Mg^2+^, as demonstrated in Figure 5A,B. This further supports our hypothesis that we measured IE enzyme activity. As mentioned earlier, IE activity is only slightly elevated by Mg^2+^ until 2 mM concentration. At higher concentrations, the enzyme activity is inhibited [12]. Similarly to this, in our experiment, in the presence of 10 mM Mn^2+^ solution, the radioactivity of [^3^H] PI was slightly elevated in 1 mM Mg^2+^ and decreased in 10 mM Mg^2+^. We need to remember that in 10 mM Mn^2+^, PIS is inhibited, hence it is further supported that it was the activity of IE that was detected. 

Phospholipid base exchange enzymes (base: phospholipid phosphatidyltransferase) catalyze the incorporation of exogenous choline, ethanolamine and serine into phospholipids. A number of phospholipids act as an endogenous substrate for choline, ethanolamine and serine exchange, and phospholipids with different fatty acid compositions are selective for different base exchange reactions. Phospholipid base exchange enzymes do not result, however, in net synthesis of phospholipid but cause significant remodeling of existing phospholipids. Although base exchange enzymes appear to represent only a minor pathway for the synthesis of most phospholipids except for phosphatidylserine, they could provide rapid energy-independent means of replenishing phospholipids, breaking down as a result of hormone or growth factor actions [16].

Inositol comes from both maternal and fetal circulations, as well as from endogenous placental synthesis, but their relative contributions are unknown [20]. As a result of inositol supplementation, it has been proven in animal experiments that the concentration of inositol in maternal and fetal serum increases [21]. This can be attributed to the transplacental transport of inositol, which was later proven in research on medullary tube closure defects [22]. In the 2000s, interest turned to myo-inositol supplementation in gestational diabetes mellitus. Recently, two meta-analyses were published on the subject [23,24]. Myo-inositol supplementation reduces the incidence of GDM and also the incidence of GDM complications (preterm delivery and gestational hypertension). Supplementation of 4 g myo-inositol/day significantly decreases the plasma glucose levels in oral glucose tolerance test (OGTT). It also decreases the need of insulin treatment and reduces the incidence of neonatal hypoglycemia [24]. Moreover, gestational diabetic women treated with myo-inositol showed better glycemic control in the 3rd trimester, and they had lower preterm birth weight [25]. It is documented that inositol improves insulin sensitivity [3,26,27]. This leads to appropriate migration and invasion of extravillous trophoblast cells. Insulin-like growth factor 1 (IGF1) plays a pivotal role in early placental development through the MAPK pathway (cytotrophoblast proliferation and syncytial formation) and phosphoinositide 3-kinase pathway (IP3) [28]. IGF1 enhances placental eNOS/NOS pathways, leading to normal trophoblast function [29]. A decrease in placental inositol concentration has been observed in GDM. This was accompanied by a reduced expression of inositol synthetizing enzymes and transporters [20]. 

In the context of inositol, the results of studies with polycystic ovary syndrome (PCOS) can also be a matter of interest. Among PCOS-affected women undergoing assisted reproductive technology, treatment with myo-inositol and high doses of chiro-inositol were shown to increase pregnancy rate and number of live births, reduce ovarian hyperstimulation syndrome and improve oocyte quality [30]; therefore, myo-inositol could improve fertility outcomes by modulating hyperandrogenism [31].

As demonstrated above, inositol has a complex and significant role in placentation, in normal pregnancy, in the treatment of gestational diabetes or in PCOS syndrome influencing fertility. Therefore, it is imperative to learn details of inositol metabolism to find alternative therapeutic ways of inositol supplementation before and during pregnancy. By detecting the IE activity in the human placenta, we provide a new pathway in the use of inositol.

## 4. Methods

There is no human inositol-exchange enzyme antibody commercially, so it was not possible to detect the enzyme activity immunologically; rather, we had to use a biochemical method.

### 4.1. Trophoblast Tissue and Microsomes

Primordial human placentae were obtained from legal social terminations after surgical evacuation of 8–10-week-old pregnancies from the Department of Obstetrics and Gynecology, Semmelweis University, Budapest, Hungary. A total number of 18 trophoblast samples were used for our experiments. Participants were healthy Caucasian women aged 21–46. No genetic testing was performed on the fetuses; however, there was no suspicion of chromosomal abnormalities or genetic conditions in the pregnancies, supported by a negative dating scan. The specific risk for Down syndrome (which is the most common chromosomopathy in the first trimester) was less than 0.3% for 15 participants and 1.1–4.0% for 3 participants. These values do not represent an increased risk for chromosomal abnormalities. Table 1 shows the data of the participants. The use of the tissues for research was approved by the Ethics Committee of the Medical Section of The Hungarian Academy of Sciences (ethical approval number: 48995-2016/EKU). The patients did not receive myo-inositol or D-chiro-inzitol therapy. Exclusion criteria were smoking, PCOS, diabetes mellitus, hypertension, known genetic disease in the family (up to immediate ancestors) and previous pregnancy complications (i.e., GDM, preeclampsia or fetal genetic disorder). We used a mixture of trophoblasts from 3-3 pregnancy terminations for our tests so that enough material was available. We mixed the samples blindly, so we did not create subgroups for the 8-, 9- and 10-week pregnancies but randomly mixed them. Since the three parallel measurements did not differ significantly from each other, the samples were considered homogeneous.

The harvested material was placed in ice-cold 0.9% NaCl, 40 mM Hepes-Na (pH 7.4), l mg/mL glucose solution and transported immediately to the biochemical laboratory for immediate separation of trophoblasts. The placenta tissue was dissected and isolated under naked eye control. The method was blindly checked occasionally by histological examinations confirming cytotrophoblasts as the predominant cell population. The trophoblasts were microdissected and minced, and the mass was measured. The minced material was used as “trophoblast tissue”.

### 4.2. Microsome Preparation

Trophoblast tissue was homogenized in 2 volumes of ice cold homogenising solution (H solution) containing 0.3 M sucrose, 0.5 mM DTT, 5 mM EDTA, 50 mM Tris/HCl at pH 7.4. The homogenate was filtered through a nylon mesh and heavy particulate material and mitochondria were sedimented at 15,000× *g* for 30 min. The supernatant was centrifuged at 100,000× *g* for 60 min to obtain microsomal pellet (MS fraction). The pellet was resuspended by homogenizing solution (H solution).

### 4.3. Crude Membrane Fraction Preparation

Trophoblast tissue was homogenized in 2 volumes of ice-cold H solution containing 0.3 M sucrose, 0.5 mM DTT, 5 mM EDTA, 50 mM Tris/HCl at pH 7.4. The homogenate was filtered through a nylon mesh, and heavy particulate material was sedimented at 2000× *g* for 10 min. The supernatant was used as crude membrane fraction (CM).

Radio-labeled [3H]inositol (50 µCi/mmol; 1.85 GBq/mmol) was obtained from ICN (Costa Mesa, CA, USA). Tris base (Tris(hydroxymethyl)aminomethane), HCl, KCl, Na_2_HPO_4_, KH_2_PO_4_, NaHCO_3_, MgSO_4_, CaCl_2_, chloroform, methanol, acetic acid, Triton X100, toluol, glucose and sucrose were purchased from REANAL (Budapest, Hungary). Ethylenediaminetetraacetic acid (EDTA), dithiothreitol (DTT), NaCl, 2,5-diphenyloxazole, p-bis [2-(5-phenyloxazolyl)]-benzene and A23187 were from Sigma Chemical Co. (Budapest, Hungary). Phenylmethylsulphonylfluoride (PMSF) and HEPES were from Calbiochem (La Jolla, CA, USA). Reagents were prepared with double-distilled deionized water.

### 4.4. Assay of Phospholipid Synthesis of Trophoblast Tissue and Microsomes

A total of 500 μL of MS fraction (0.48–1.4 mg protein/point) or 300–500 mg trophoblast tissue was placed in the shaking incubator with PO_4_^3−^-and Ca^2+^-free Hank solution and 0.5–2.5 μCi/mL [3H]inositol (10–50 nmol) in 2 mL final volume for 30–180 min at 37 °C. During our investigations, we did not have the opportunity to measure the tissue inositol concentration. Endogenous inositol reduces the radioactivity measured by the incorporation of [3H]inositol in the tissue but does not affect the regulation of PIS and IE and does not modify the observed results. The amount of trophoblast in each case was limited, so different masses of the tissue were used for the measurement. In our preliminary experiments, using 100–600 mg of tissue, we found that the specific enzyme activity (cpm [3H]inositol/mg protein) does not change, so this does not modify the result. When preparing microsomes, water-soluble small-sized substances, including endogenous inositol, were removed [32].

### 4.5. Trophoblast Tissue

A total of 300–500 mg minced trophoblast tissue (in the same experiment always identical amounts) was incubated in 2.0 mL Hank medium without Ca^2+^ and PO_4_^3−^ buffered with 40 mM Hepes/Na (pH = 7.4). The incubations were conducted in open vials shaken continuously in a 37 °C water bath for the time intervals specified. The reaction was terminated by 5 mL ice cold 0.1 M acetic acid, 0.9% NaCl solution. Lipids were extracted using the method of Tóth et al. [33]. Samples were centrifuged at 2000× *g* for 10 min, and the precipitated tissue was collected and suspended in 0.6 mL 0.9% NaCl and 3.0 mL methanol and homogenized with glass potter. The potter was washed in 1.2 mL methanol. The phospholipids were extracted in the chloroform (lower) phases, and these were collected. The remaining phospholipid from the aqueous phase was purified with chloroform. The compound was stirred with 1.5 mL chloroform then centrifuged at 2000× *g* for 10 min. The under phase was stirred with 0.6 mL 0.9% NaCl, 0.1 mL 1 N HCl, 3.0 mL methanol for 30 s and added to and stirred with 1.5 mL chloroform. The supernatants were collected. The collected supernatants were washed 3 times in 3 mL washing solution (chloroform:methanol:2 M KCl = 3:48:47 (*v*/*v*%)), centrifuged again, and the supernatant was removed. The evaporation residue was solved in 5 mL chloroform:methanol = 1:1 (*v*/*v*). A total of 4x1 mL was evaporated, and radioactivity was measured in 2 mL measuring solution (Triton X100:toluol = 1:2 (*v*/*v*), 3.33 g/L, 2.5-diphenyloxazole, 0.033 g/L p-Bis [2-(5-phenyloxazolyl)]-benzene).

### 4.6. Microsomes

The 0.48–1.40 mg MS protein/sample (in the same experiment always identical amounts) was incubated in 2.0 mL Hank medium without Ca^2+^ and PO_4_^3−^ buffered with 40 mM Hepes/Na (pH = 7.4). The incubation was conducted in open vials shaken continuously in a 37 °C water bath for the time intervals specified. The process was stopped by 4 mL methanol:chloroform:6 N HCl = 20:10:0.7 (*v*/*v*) solution. The samples were centrifuged at 3000 rpm for 5 min, and the supernatant was mixed with 3 mL chloroform and centrifuged at 3000 rpm for 5 min. The supernatant was removed and washed 3 times in 3 mL washing solution (chloroform:methanol:2 M KCl = 3:48:47 (*v*/*v*%)) and centrifuged again, and then the supernatant was removed. The last under phase was evaporated in mini vials and suspended in 5 mL measuring solution and was measured.

### 4.7. Protein Determinations

Protein amount was measured by the conventional method of Lowry et al. using bovine serum albumin as standard [34].

### 4.8. Data Analysis

The results were statistically evaluated by the GraphPAD InStat 1.2 program. The difference was considered to be significant if the “*p*” value was <0.05 (Bonferroni multiple comparison). The graphic presentations were made with the help of Microsoft Office PowerPoint 2016 and Microsoft Office Excel 2016 programs.

## 5. Conclusions

Our paper demonstrates that in addition to PIS, there is another pathway (IE) of PI synthesis in the placenta. PI derivates play a pivotal role in the growth signaling pathway, which is a very intensively researched area, but still only few results exist in the literature about the actual biochemistry of PI synthesis. Furthermore, during inositol supplementation of pregnant women, inositol can be used not only in the PIS pathway but also alternatively in the IE pathway, so in cases of inositol deficiency, the two pathways should be examined together. During our experiments, we confirmed the activity of IE in the primordial placenta. Accurate measurement of the difference in IE activity between primordial placenta and mature placenta will require further studies.

## Figures and Tables

**Figure 1 ijms-25-03436-f001:**
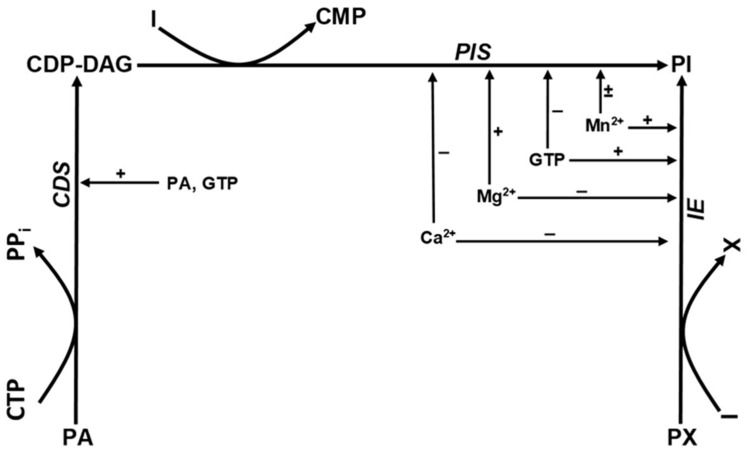
Regulation of phosphatidylinositol synthesis.

**Figure 2 ijms-25-03436-f002:**
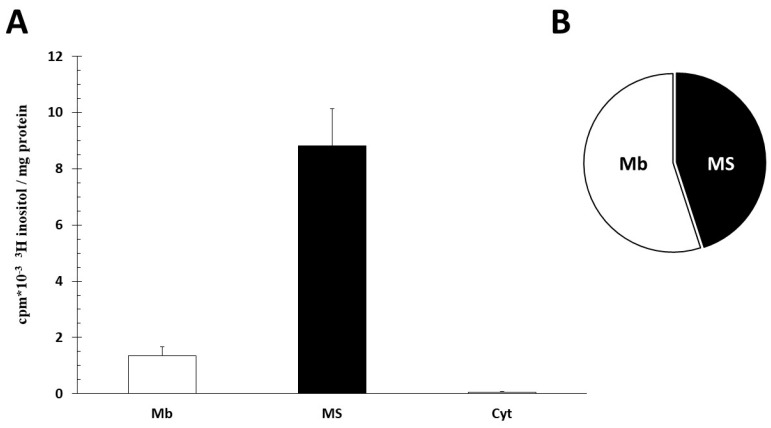
Intracellular distribution of inositol incorporation. (**A**) Specific radioactivity of crude membrane (Mb), microsomes (MS) and cytosolic (Cyt) [3H] inositol incorporation in phosphatidyl inositol. Samples were incubated for 60 min at 37 °C, Ph = 7.4, 2.5 μCi/mL [3H] inositol. (**B**) Distribution of total radioactivity. Total activity was calculated from protein concentration and tissue volume. (*n* = 6 ± SD).

**Figure 3 ijms-25-03436-f003:**
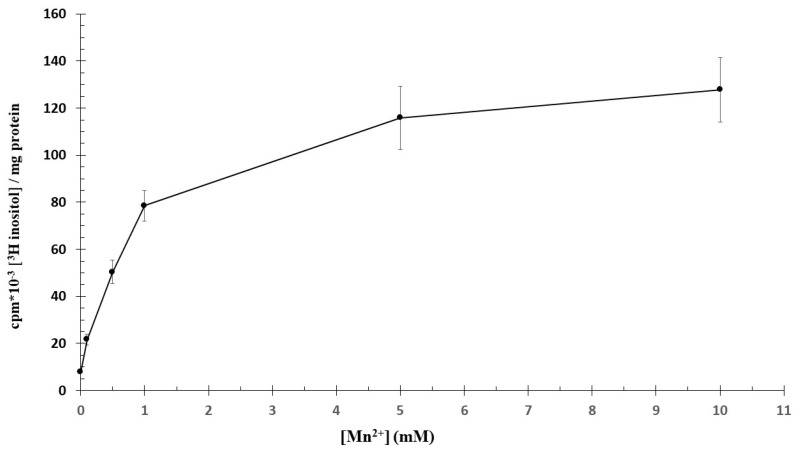
[3H]inositol incorporation of human primordial placenta with Mn^2+.^400 mg trophoblast tissue was incubated for 60 min at 37 °C, pH = 7.4, 2.5 μCi/mL [3H]inositol with Mn^2+^. (*n* = 6 ± SD).

**Figure 4 ijms-25-03436-f004:**
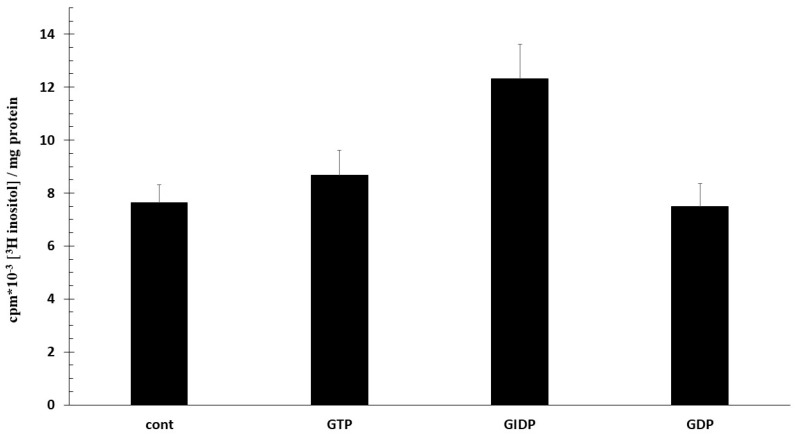
Effect of GTP, GIDP and GDP in phosphatidylinositol synthesis. Trophoblast microsomes were incubated for 60 min at 37 °C with 10 mM Mg^2+^ in pH = 7.4, 2.5 μCi/mL [3H]inositol; 0.1 mM GTP, GIDP, GDP were added at 0 and 30 min. (*n* = 3 ± SD).

**Figure 5 ijms-25-03436-f005:**
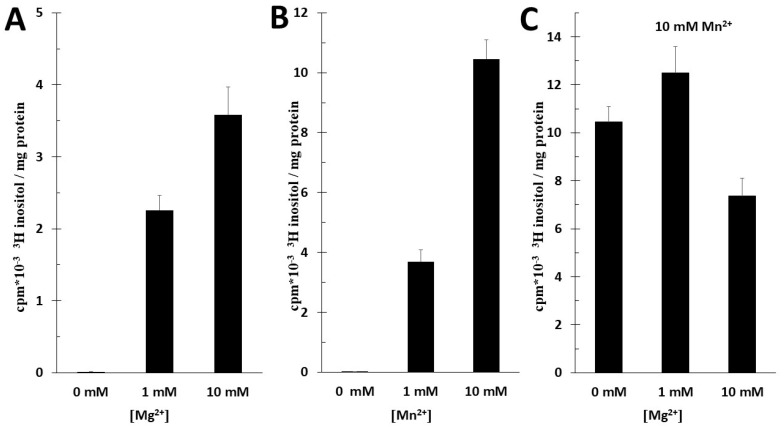
(**A**) Incorporation of [3H]inositol in phosphatidylinositol in microsomes of trophoblast with Mg^2+^. Microsomes were incubated with 2.5 μCi/mL [3H]inositol for 60 min at 37 °C in pH = 7.4, with Mg^2+^. (**B**) Incorporation of [3H]inositol in phosphatidylinositol in microsomes of trophoblast with Mn^2+^. Microsomes were incubated with 2.5 μCi/mL [3H]inositol in pH = 7.4 for 60 min at 37 °C with Mn^2+^. (**C**) Incorporation of [3H]inositol in phosphatidylinositol in microsomes of trophoblast with Mg^2+^ and 10 mM Mn^2+^. Microsomes were incubated with 2.5 μCi/mL [3H]inositol in pH = 7.4 for 60 min at 37 °C with Mg^2+^ and 10 mM Mn^2+^. (*n* = 6 ± SD).

**Table 1 ijms-25-03436-t001:** Clinical data mean ± SD; (*n* = 18).

Maternal age (years)	30.61 ± 7.80
BMI (kg/m^2^)	23.64 ± 4.43
Gestational age (week)	8.89 ± 0.66
Number of pregnancies	3.22 ± 2.29

## Data Availability

Data are contained within the article.

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
