# Peer review of "Inositol-Exchange Activity in Human Primordial Placenta"

_ijms, 2024, doi:10.3390/ijms25063436_

Round 1
Reviewer 1 Report
Comments and Suggestions for Authors
Dear authors,
Thanks for the paper that is submitted, which covers an interesting topic. I think this article attempts to explain the activity of de Inositol Exchange enzyme and, furthermore, the physiological properties of this enzyme and phosphatidylinositol synthase.
The article is well written and flows easily. I have major comments and edits that I would like to suggest to the authors:
Abstract: The extension is too long and does not follow the requirements of the journal.
Introduction section:
Lines 54-56/Lines 345-346: In scientific writing, one sentence should not conform to a paragraph.
Method Section: Primordial human placentae were obtained from legal social terminations. My question is: Did these abortions have a high suspicion of genetic alterations (chromosomopathies) or malformations? If so, the results would be biased.
Figure 5: I suggest to reduce the size of the figure
I suggest introducing a Conclusion section.
I suggest to complete the Author Contributions; Funding; Institutional Review Board Statement.
Please, check the journal s recommendations, in the citations, Tables and Figures. All references do not follow the ACS style guide.
Author Response
Dear Reviewer,
Firstly, we would like to thank your helpful critiques which we really do believe that helped us to raise the quality of our work. We have revised our manuscript in order to match your recommendations.
Balázs Mészáros,
on behalf of the authors

Reviewer 2 Report
Comments and Suggestions for Authors
Interesting work that explores the important role of the possible synthesis of phosphatidylinositol in the placenta and the possible clinical implications of the administration of inositol during pregnancy
Comments
-The authors should provide information on the patients studied. It is known that inositol is reduced in PCOS and diabetes, pathologies that are very common in women and are associated with reduced plasma concentrations of inositol
-Specify whether abortion at 8 and 10 weeks could alter the results. In particular, whether there was a relationship between the study results and the weeks of pregnancy at the time of the abortion.
-Line 138 explain SIP
-Explain whether the quantity of cytosol and microtomes was always the same in the various patients and whether the trophoblastic tissue of each patient has been evaluated separately
-Line 169: 300-500 minced trophoblast tissue (in the same experiment always identical amounts) explain whether this mass is proportionally the same compared to the weight of all the material obtained
-Explain whether there is a possible interference of different quantities of minced trophoblast tissue on the results of the various experiments
-Results: does the synthesis of placental inositol derive from circulating inositol? Were any of the patients taking myo or chiroinositol?
-line 214 explain the dose of 3Hinositol contained in 2.5& Ci
-In all figures explain how many experiments were done and whether each experiment was done on a single trophoblast or whether multiple cases were mixed. The number of cases is mentioned only in fig 2
-Explain in the introduction or discussion the difference between primordial placenta and mature placenta in terms of inositol and OE activity
-In the discussion it is reported that inositol is produced both in the maternal and fatal circulation explain whether the circulating inositol passes the placenta and whether the placental concentration is related to the systemic one
Author Response
-The authors should provide information on the patients studied. It is known that inositol is reduced in
PCOS and diabetes, pathologies that are very common in women and are associated with reduced
plasma concentrations of inositol – lines 254-263 and Table 1
-Specify whether abortion at 8 and 10 weeks could alter the results. In particular, whether there was a
relationship between the study results and the weeks of pregnancy at the time of the abortion. – lines
254-269
-Line 138 explain SIP – corrected (PIS)
-Explain whether the quantity of cytosol and microtomes was always the same in the various patients
and whether the trophoblastic tissue of each patient has been evaluated separately Lines 264-266
-Line 169: 300-500 minced trophoblast tissue (in the same experiment always identical amounts)
explain whether this mass is proportionally the same compared to the weight of all the material
obtained Lines 263-264
-Explain whether there is a possible interference of different quantities of minced trophoblast tissue on
the results of the various experiments Lines 266-267 and 300-311
-Results: does the synthesis of placental inositol derive from circulating inositol? Lines 216-220. Were
any of the patients taking myo or chiroinositol? Line 260-261
-line 214 explain the dose of 3Hinositol contained in 2.5& Ci – corrected in line 111
-In all figures explain how many experiments were done and whether each experiment was done on a
single trophoblast or whether multiple cases were mixed. Lines 256-264 The number of cases is
mentioned only in fig 2 – corrected. Lines: 142, 153, 170
-Explain in the introduction or discussion the difference between primordial placenta and mature
placenta in terms of inositol and IE activity – lines 358-360
-In the discussion it is reported that inositol is produced both in the maternal and fatal circulation
explain whether the circulating inositol passes the placenta and whether the placental concentration is
related to the systemic one Lines 216-220

Round 2
Reviewer 1 Report
Comments and Suggestions for Authors
Dear authors,
Thank you for taking the time to address comments on the manuscript. The manuscript has been greatly improved, but there are still some concerns about some issues.
I think the Materials and Methods section should be placed after the Introduction section. So, the result section remains with the number 3.
Method Section: Primordial human placentae were obtained from legal social terminations. My question is: Did these abortions have a high suspicion of genetic alterations (chromosomopathies) or malformations? If so, the results would be biased - Lines 254-263
There is any reference in lines 254-263 (as it is salid in the cover letter) related to the origin of trophoblast tissue.
I would appreciate for these comments to be taken into consideration.
Author Response
Dear Reviewer,
We are grateful for your comments, which play a big role in improving the quality of our manuscript. In our newest version, we have attempted to provide detailed answers.
Comment: I think the Materials and Methods section should be placed after the Introduction section. So, the result section remains with the number 3.
Answer: According to the journal’s guidelines, the correct order of sections is as follows: 1. Introductions 2.Results 3.Discussions 4. Materials and Methods 5. Conclusions
2. Comment: Method Section: Primordial human placentae were obtained from legal social terminations. My question is: Did these abortions have a high suspicion of genetic alterations (chromosomopathies) or malformations? If so, the results would be biased - Lines 254-263
Answer: line 258-264
Participants were healthy Caucasian women aged 21-46. No genetic testing was performed on the fetuses, however, there was no suspicion of chromosomal abnormalities or genetic conditions in the pregnancies, supported by a negative dating scan. The specific risk for Down syndrome (which is the most common chromosomopathie in the first trimester) was less than 0,3% for 15 participants and 1,1-4,0 % for 3 participants. These values are not represent an increased risk for chromosomal abnormalities.
Remark: Our ethics license did not cover genetic testing. We did not think that it might be necesarry, because the participants were healthy women. Cases with positive anamnesis, or genetic disease during previous pregnancy were excluded.
- Comment: There is any reference in lines 254-263 (as it is salid in the cover letter) related to the origin of trophoblast tissue.
Answer: line 258-259
Participants were healthy Caucasian women aged 21-46.
Remark: If we misunderstood the question, please specify what data we should add to our answer.
Yours Faithfully,
Bence Géza Kovács